# Immunological Roles of *TmToll-2* in Response to *Escherichia coli* Systemic Infection in *Tenebrio molitor*

**DOI:** 10.3390/ijms232214490

**Published:** 2022-11-21

**Authors:** Maryam Ali Mohammadie Kojour, Ho Am Jang, Yong Seok Lee, Yong Hun Jo, Yeon Soo Han

**Affiliations:** 1Department of Applied Biology, Institute of Environmentally-Friendly Agriculture (IEFA), College of Agriculture and Life Sciences, Chonnam National University, Gwangju 61186, Republic of Korea; 2Department of Biology, College of Natural Sciences, Soonchunhyang University, Asan 31538, Republic of Korea

**Keywords:** Toll-2, *Tenebrio molitor*, microbial infection, antimicrobial peptides, RNAi technology

## Abstract

The antimicrobial roles of Toll-like receptors have been mainly identified in mammalian models and *Drosophila*. However, its immunological function in other insects has yet to be fully clarified. Here, we determined the innate immune response involvement of *TmToll-2* encountering Gram-negative, Gram-positive, and fungal infection. Our data revealed that *TmToll-2* expression could be induced by *Escherichia coli*, *Staphylococcus aureus*, and *Candida albicans* infections in the fat bodies, gut, Malpighian tubules, and hemolymph of *Tenebrio molitor* young larvae. However, *TmToll-2* silencing via RNAi technology revealed that sole *E. coli* systemic infection caused mortality in the double-strand RNA *TmToll-2*-injected group compared with that in the control group. Further investigation indicated that in the absence of *TmToll-2*, the final effector of Toll signaling pathway, antimicrobial peptide (AMP) genes and relevant transcription factors were significantly downregulated, mainly *E. coli* post-insult. We showed that the expression of all AMP genes was suppressed in the main immune organ of insects, namely, fat bodies, in silenced individuals, while the relevant expressions were not affected after fungal infection. Thus, our research revealed the immunological roles of *TmToll-2* in different organs of *T. molitor* in response to pathogenic insults.

## 1. Introduction

Toll and Toll-like receptors (TLRs) are a conserved family of pattern recognition receptors (PRRs) initially identified for their role in dorso-ventral axis formation in *Drosophila* embryos [1]. The TLR/nuclear factor-kappaB (NF-κB) pathway activates immune responses in *Drosophila* [2,3]. Since the first Toll protein (Toll1) and eight other Toll receptors (18-Wheeler and Toll2-9) were characterized in *Drosophila*, 10 TLRs in human (TLR1-10) and 13 TLRs in mouse (TLR1-13) have been identified [4]. TLR activation and function can vary between vertebrates and invertebrates. In vertebrates, the Toll pathway is activated by direct interaction with pathogen-associated molecular patterns (PAMPs) [5]. Whereas, in invertebrates, Toll signaling is initiated by binding to the cytokine-like molecule Spätzle (Spz) [4]. Thus, it is proposed as evolutionary conversion in Toll/TLR immune functions [6]. Although the Toll pathway in invertebrates has immunity and developmental activities, mammalian TLRs are simply involved in immunity and have no developmental roles [7].

The structural properties of insect and mammalian TLRs share analogies and differences [8]. Insect Toll and vertebrate TLRs are characterized by leucine-rich repeat (LRR) domains containing cysteine-rich motifs in the extracellular domain, a single-pass transmembrane domain, and a cytoplasmic Toll/Interleukin-1 receptor (TIR) domain [9]. Although the TIR domain has similarities to that of the IL-1 receptor family, unlike the intracellular domain of Tolls, IL-1 receptors consist of an immunoglobulin-like domain [10]. Conversely, mammalian TLRs contain a single cysteine-rich cluster at the C-terminal end of LRRs, except Toll9; other *Drosophila* Tolls hold cysteine-rich clusters at either, or both, the N- and C-terminal ends of LRRs [11]. Likewise, *Drosophila* Toll9 was shown to be the closest TLR to vertebrate counterparts, suggesting functional similarities among these receptors in different clades [9].

Similar to its receptor structure, the Toll signaling pathway in insects is divided into three steps: (1) extracellular recognition of pathogenic antigens, including but not limited to Lys-type peptidoglycan (Lys-PGN) of Gram-positive bacterial cell walls, β-glucans of fungi, and virulence factors, by PGN recognition proteins (PGRPs) and glucan-binding proteins (GNBPs); (2) activation of a distinct proteolytic cascade from the developmental activation of Toll signaling, leading to the cleavage and activation of the zymogen Spätzle and binding of a cleaved ligand to the extracellular domain of TLRs; and (3) dimerization of the receptor cytosolic domain, TIR, activation of a phosphorylation cascade of an intracellular signaling cascade, and expression of effector molecules [12,13]. After the activation of PGRP-SA/GNBP1 and GNBP3 by Lys-PGN of Gram-positive bacteria cell walls and β-1,3-glucans of yeast and some fungi, respectively, a cascade of CLIP-domain zymogens mediate signal amplification [12]. In addition to the serine protease cascade, modular serine protease (ModSP) and Grass in turn activate the Spätzle-processing enzyme (SPE) [14,15]. The protease Persephone is likely involved in this proteolytic cascade activation [13]. However, the precise mechanism by which protease directly cleaves and activates SPE is yet to be clarified [16]. The interaction of the mature active C-terminal C-106 domain of Spz leads to the dimerization of intracytoplasmic TIR domains, which in turn, interact with distinct death-domain-containing proteins [2]. Upon the binding of processed TIR to the death-domain-containing adaptor molecules, Myeloid differentiation primary response 88 (MyD88), it is associated with the death domain of Tube [17]. Through a bifunctional death domain, Tube recruits the Ser/Thr kinase Pelle [18]. Subsequently, NF-κB transcription factors are released from their inhibitors through the K48 ubiquitination and degradation of Cactus. Nevertheless, the Cactus kinase is not identified [12]. Freed Rel transcriptional factors, Dorsal-related immunity factor (Dif), and Dorsal are then translocated into the nucleus, bind to κB-response elements, and induce the transactivation of antimicrobial peptides (AMPs), which are the hallmark of insect innate immune responses [12,15,19].

Studies on immune responses of arthropods have focused on dipterans [16]. Moreover, a relatively large insect model should be established for comprehensive and accurate biochemical studies (sufficient hemolymph sampling) by using species belonging to different taxonomic groups [6]. To this extent, Toll signaling pathway and their components in *T. molitor* were investigated in response to various pathogenic sources, including Gram-positive and Gram-negative bacteria and fungi. Seven *Toll* genes (*TmToll-2*, *-3*, *-6*, -*7*, -*8*, *-9*, and *-10*) were identified in *T. molitor*. However, the functional importance of these isoforms, except *TmToll-2*, is poorly understood [20]. In the present study, we explored the immunological importance of *TmToll-2* against microbial infection. Our study revealed that *Tm*TLR2 is important for immune-mediating responses to Gram-negative *E. coli* (Appendix A). Our findings might provide a basis for developing possible therapeutic strategies against human pathogens.

## 2. Results

### 2.1. Sequence Analysis of TmToll-2

In this study, a Toll-2 homolog from *T. molitor* (*TmToll-2*, Accession number: OP566501) was identified by an EST and RNA-seq search by using the *T. castaneum* protein sequence as a query. A 2510 bp open reading frame (ORF) encoded a protein of 834 amino acids. Phylogenetic analysis based on the full-length amino acid sequences of *Tm*Toll-2 and other insect Toll receptors indicated that it clustered with Toll-2 proteins from *T. castaneum*, *Sitophilus oryzae*, *Manduca sexta*, *Vanessa cardui*, *Bicyclus anynana*, *Galleria mellonella*, *Neodiprion fabricii*, *Nilaparvata lugens*, *Homalodisca vitripennis*, TLR4 from *Bombyx mori*, Toll-9, isoform C from *Drosophila melanogaster*, and TLR Tollo from *Anopheles arabiensis*; and *Mus musculus* TLR2, the isoform X1, was used as the outgroup (Figure 1). Phylogenetic analysis showed that *Tm*Toll-2 in the order Coleoptera formed a group with other isoforms of Toll-2 from *T. castaneum* and *S. oryzae*.

### 2.2. Developmental and Tissue Expression of TmToll-2

The mRNA expression of *TmToll-2* was evaluated through qRT-PCR at different developmental stages (Figure 2A) and in different tissues of late-instar larvae (Figure 2B) and adults (Figure 2C). Our data illustrated the developmental and innate immunity accepts of TLR receptors and revealed that the highest expression pattern of *TmToll-2* occurred in 1-day-old adults and the embryonic stage in a descending order. In young larvae and adults of *T. molitor*, the gene expression of *TmToll-2* was the highest in gut tissues but negligible in other tissues (less than 0.1-fold change).

### 2.3. Temporal Expression of TmToll-2 Post-Systemic Infection

The mRNA expression level in *T. molitor* young larvae was measured and examined post-systemic infection with *E. coli*, *S. aureus*, and *C. albicans* in the MTs (Figure 3A), gut (Figure 3B), hemolymph (Figure 3C), and fat bodies (Figure 3D) 3, 6, 9, 12, and 24 h after injection to determine the immunological roles of *TmToll-2* in deflecting pathogenic attacks. PBS was used as the control. The results showed that *TmToll-2* mRNA expression was upregulated in response to all infectious sources and varied in tissue- and time-dependent manner. The upregulation of the *TmToll-2* gene expression was the highest in the hemolymph and MTs in a descending order against *C. albicans* 9 h post infection (up to 350- and 90-fold, respectively). Moreover, the expression of *TmToll-2* was induced by 150- and 100-fold against *S. aureus* and *E. coli*, respectively, because of the involvement of the Toll signaling pathway in the recognition of Gram-positive and Gram-negative invasion. However, the minor gut expression of *TmToll-2* was induced mainly after *E. coli* infection by approximately 3-fold and *S. aureus* by approximately 2-fold but not after *C. albicans* infection. Likewise, *TmToll-2* expression was relatively low in fat bodies in response to all microorganisms. 

### 2.4. Effect of TmToll-2 RNAi on T. molitor Larval Survival

After we observed the *TmToll-2* mRNA induction post systemic infection, we further investigated the innate immune responses in *T. molitor* larvae treated with ds*TmToll-2* RNAi. Initially, we checked the survival rates of the infected *TmToll-2*-silenced larvae compared with those of ds*TmVer*-treated larvae as the control group. Furthermore, 4 days after ds*TmToll-2* RNAi injection, *TmToll-2* mRNA levels were decreased by 77% (Figure 4A), confirming the efficiency of RNAi. Subsequently, *T. molitor* larvae were monitored during 10 days after *E. coli* (Figure 4B), *S. aureus* (Figure 4C), and *C. albicans* (Figure 4D) injection. We did not report either significant or meaningful mortality rate between *S. aureus*- and *C. albicans*-infected ds*TmToll-2*- and ds*TmVer* RNAi-treated groups, except *E. coli*-infected larvae. Therefore, a high survivability rate was reported after non-entomopathogen infections.

### 2.5. Effect of TmToll-2 Gene Silencing on the Expression of Antimicrobial Peptide and NF-κB Genes

An interesting result of *TmToll-2* gene silencing indicated that the absence of this gene did not affect the survivability of *T. molitor* larvae following Gram-positive and fungal infections. However, the *TmToll-2* mRNA expression significantly upregulated in hemolymph and Malpighian tubules in response to bacterial and fungal infections. Therefore, to verify these observations, we checked the hallmark of *T. molitor* innate immunity and the regulation of 15 AMP genes: *TmTenecin-1*, *-2*, *-3*, and *-4* (*TmTene1*, *2*, *3*, and *4*); *TmAttacin-1a*, *-1b*, and *-2* (*TmAtt1a*, *1b*, and *2*); *TmDefensin* (*TmDef*); *TmDefensin-like* (*TmDef-like*); *TmColeoptericin-A*, *-B*, and *-C* (*TmColeA*, *B*, and *C*); *TmCecropin-2* (*TmCec-2*); and *TmThaumatin like protein-1* and *-2* (*TmTLP1* and *2*) in *TmToll-2*-silenced larvae challenged with *E. coli*, *S. aureus*, and *C. albicans*. We also confirmed the *TmToll-2* knockdown efficiency. In fat bodies, which are the main immune organ of insects, 13 out of 15 AMP genes were downregulated in response to *E. coli*, and 14 out of 15 AMP genes were downregulated in response to *S. aureus* (Figure 5 A–O). In accordance with previous reports of AMP production in *T. molitor* [20,21,22], we showed that *TmDef*, *TmDef-like*, and *TmCec2* are involved in immune responses against Gram-positive, Gram-negative, and fungal infections (Figure 5E,F). Likewise, the downregulation of *TmTLP2* following *C. albicans* infection in *TmToll-2*-silenced larvae presented the analogy of previous findings (Figure 5O). In the gut (Figure 6A–O), eight AMP genes were downregulated in *TmToll-2*-silenced larvae after *E. coli*, *S. aureus*, and *C. albicans* infections. However, the main effect of *TmToll-2* gene silencing in the gut could be observed after *E. coli* infection, particularly *TmTene2* (5000-fold), *TmTene4* (2000-fold), *TmColeA* (2000-fold), and *TmColeC* (3500-fold), compared with that in ds*TmVer*-treated larvae (Figure 6B,D,H,J). Similarly, in Malpighian tubules (Figure 7A–O), the superior negative regulation in most AMP genes, including *TmTene1* (Figure 7A), *TmTene4* (Figure 7D), *TmDef-like* (Figure 7F), *TmCec2* (Figure 7G), *TmColeA* (Figure 7H), *TmColeB* (Figure 7I), *TmColeC* (Figure 7J), *TmAtt1a* (Figure 7K), and *TmAtt1b* (Figure 7L), was observed after *E. coli* infection. Hence, *S. aureus* injection in ds*TmToll-2* RNAi-treated larvae remarkably downregulated the expression levels of *TmTene1* (Figure 7A), *TmTene4* (Figure 7D), *TmDef-like* (Figure 7F), *TmColeA* (Figure 7H), *TmColeB* (Figure 7I), *TmColeC* (Figure 7J), *TmAtt1a* (Figure 7K), and *TmAtt1b* (Figure 7L). Additionally, the AMP genes were marginally negatively regulated after *C. albicans* in the *TmToll-2*-silenced group. Interestingly, ds*TmToll-2* RNAi increased the mRNA levels of some AMPs in response to pathogens in the whole body samples (Figure 8A–O), gut, and Malpighian tubules, precisely the levels of *Cecropin*, *Coleoptericin*, *Attacin*, *Thaumatin-like protein*, and *Tencin* families (Figure 6A,E–G,I–M and Figure 7B,E,M–O). Unexpectedly, seven AMP genes in the whole body samples were mostly upregulated in response to all infections, explaining the high survivability rate in response to invasions (Figure 8A–O).

In accordance with the same protocol used to evaluate the expression of AMP genes after *TmToll-2* knockdown, we further examined the NF-kB pathway genes *TmDorX1*, *TmDorX2*, and *TmRelish* in the whole body, fat bodies, gut, and Malpighian tubules (Figure 9A–D) to determine the exact transcription factors downstream of the main signaling pathways responsible for AMP production. Consistent with previous findings and AMP expression levels, the results showed that the expression level of *TmDorx2*, downstream of the Toll signaling pathway, decreased following *E. coli* and *S. aureus* infection in all the examined tissues, whereas the level of *TmRelish*, downstream of Imd signaling, was not strongly affected [23,24,25,26]. Additionally, the comparison of the expression pattern of *TmDorX1* in different tissues explained the main negative and positive regulation of AMP gene expression; in the whole body, *TmDorX1* was positively regulated (Figure 9A). Conversely, in the fat bodies and gut, the NF-kB gene expression was negatively regulated (Figure 9B,C).

## 3. Discussion

The Toll signaling pathway is important for the development and immunity of insects in their life. However, intensive studies have described the roles of TLRs in infectious diseases and innate immunity; furthermore, the current understanding of these receptors and relevant signaling is chiefly limited to mammalian and some invertebrate models such as *Drosophila*, mosquitoes, moths, beetles, and shrimp. These receptors are evolutionarily conserved among all living organisms from mammals to plants [6]. Homologies between toll receptors in vertebrates and invertebrates can be characterized by TIR and LRR domains [27]. Except for Toll-1 in *D. melanogaster* [28], the specific functions, including their immunological roles, of other proteins within this family remain elusive.

With economic importance, *T. molitor* was developed into a distinctive immune study model. We identified seven Toll genes in *T. molitor*. *TmToll-7* roles in the innate immune response following Gram-negative *E. coli* infection through AMP production was reported [20]. Here, we attempted to reveal further roles in the TLR family by examining *TmToll-2* activity in warding off pathogenic invasions via RNAi technology.

Initially, our phylogenic analysis indicated that the Toll-2 cluster in insects is distinguished from mammalian TLRs, suggesting distinct evolutionary events and consequently different co-players and functions [29]. In addition to their roles in dorso-ventral axis formation, TLRs have a dynamic expression pattern throughout the development of *Drosophila* [30]. Moreover, Toll-2, -6, -7, and -8 in *T. castaneum*, which forms a group with *T. molitor*, with 62% amino acid identities in our phylogenic analysis, share a common characteristic-encoding more LRRs, and *Drosophila* long Tolls function in embryonic development [9]. Accordingly, *TmToll-2* expression patterns vary in an insect’s life; in addition to the embryonic stage, the highest expression in 1-day-old adults highlights the TLR significance in the entire development of insects. The early adult stage expression of *TmToll-2* can be attributed to compartment boundary restrictions, which are mediated by Toll-1 and -2 as adhesion molecules in *Drosophila* [31]. Moreover, the highest *TmToll-2* expression in the gut of *T. molitor* larvae and adults in tissue-specific gene expression experiments is supported by former reports, which show that Imd signaling within an insect’s gut is not the solo player in an innate immunity match; instead, a possible crosstalk exists between Imd and Toll signaling pathways [32,33].

Toll signaling in *T. molitor* can be activated after various microbial challenges, including but not limited to Gram-positive, Gram-negative, and fungal infections [20,25,26]. The DAP-type peptidoglycan (DAP-PGN) of Gram-negative bacteria can be sensed by *Tenebrio* recognition protein PGRP-SA and induces activation [34]. Here, we reported that *S. aureus*, *C. albicans*, and *E. coli* induce *TmToll-2* mRNA expression, and the relevant expression was distinguished in the hemolymph of *T. molitor* larvae. Previous studies showed that the Toll-dependent wound healing activity is initiated by the influx of extracellular calcium ions in epidermal cells in the wound site of *Drosophila*, which eventually triggers Spätzle cleavage and activation [35]. Additionally, transglutaminase activity and calcium ions from injury in insects mediates the prophenoloxidase-activating (proPO) system [8]. The same protease cascade leading to the PO activity can mediate Spätzle cleavage and Toll pathway-dependent AMP production in *M. sexta* [13,36]. Consequently, the extensive induction of *TmToll-2* in the hemolymph samples of the pathogen-injected groups could be triggered by PO activity, the injection site of epidermal cells with oxidative stress, and proteolytic cascade activation following microbial sensation by Toll-related PRRs. Because of the constant hemolymph flow into Malpighian tubules [37], the expression pattern of *TmToll-2* in this tissue was also relatively high.

*TmToll-2*-silenced larvae were more susceptible to Gram-negative infection than to the controls. In accordance with the survivability result, AMP expressions showed to be in favor of *T. molitor* larvae after *E. coli* in the *T. molitor* vermilion (*TmVer*) double-strand RNA (ds*TmVer*) injected group. The negative regulation of all AMPs in the fat bodies of insects following Gram-negative bacterial infection was consistent with our previous report on *TmToll-7*, in which AMPs downregulated in the silenced larvae following *E. coli* infections [20]. Moreover, we observed that the mRNA expression levels of *TmTen2*, *TmTen4*, *TmColeA*, *TmColeC*, *TmCec2*, and *TmTLP1* were negatively regulated in *TmToll-2*-silenced individuals. The secretion of cecropin and coleoptericins is dependent on Imd signaling in *Drosophila* and weevil *Sitophilus*, respectively [38,39]. Additionally, the gut multifunction in insect metabolism and immunity is extensively described [40]; conversely, Imd and Toll signaling pathways play major roles in insects’ innate immunity, and the Toll pathway in *Drosophila* does not have a role in gut immunity [41]. Hence, reactive oxygen species (ROS) production and Imd signaling function as a homeostasis regulator of the gut microbiota in *Drosophila* and red palm weevil (RPW) [42,43]. Similar to our former studies in *T. molitor* and RPW [14], the present study exhibited that *Tm*Toll-2 could mediate immunity within the gut by regulating the AMP production, confirming that (*i*) the Imd signaling pathway mainly functions in immunity within the gut [44,45,46]; (*ii*) Gram-negative bacteria are recognized by PRRs of Toll signaling [47,48]; (*iii*) and possible cross-talks existed between Toll and Imd signaling pathways [33,49]. Interestingly, in *Drosophila*, AMP expression in fat bodies can be regulated through ROS signaling in the gut, and hemocytes serve as a signal-relaying organ between the gut and fat bodies after oral infection [50]. This finding can explain why *T. molitor* larval fat bodies invest the least energy on the expression of the *TmToll-2*, because the signaling pathway, its relevant transcription factor, and AMPs as final effectors can be also activated by other organs after the infections. Surprisingly, the positive regulation of the mRNA expression of APMs in the whole body samples suggested that major immune organs, such as fat bodies, gut, and Malpighian tubules, affect the total AMP production in *T. molitor* larva; other tissues and relative cells, such as the integument and hemocytes, are vital players in the overall immunity of insects [51]. Furthermore, the mRNA production of AMPs in the whole body can explain a high survival ratio post systemic infection by different pathogens. Studies on NF-κB function under pathological conditions revealed that the mRNA levels of several AMPs are regulated by *TmRelish* (downstream of Imd signaling) and *TmDorX2* (downstream of Toll signaling) pathways in the fat body [45,52]. Accordingly, our NF-κB results were consistent with the mRNA expression of AMPs. This result suggested that the negative relationship between *TmDorX2* and *TmRelish* is responsible for the downregulation of AMP genes in dissected tissues. Moreover, in whole body samples, *TmDorX1* is positively regulated, indicating the upregulation of AMP genes in the relevant samples.

Overall, our molecular analysis provided insights into the innate immunity of *T. molitor*. We illustrated that *TmToll-2* regulates antimicrobial activities in epithelial tissues such as Malpighian tubules and the gut, in addition to fat bodies. *TmToll-2* is activated by Gram-positive and Gram-negative bacteria in *T. molitor*, similar to *B. mori* but in contrast to some insects such as *M. sexta* and *D. melanogaster* [2]. In addition, the activation of AMP genes was nonspecific after *E. coli*, *S. aureus*, and *C. albicans* challenges. Further comprehensive studies of possible ligands and simulators of TLRs and possible cross-talks of this signaling with other pathways should provide a clear perspective of the underlying mechanisms involved in innate immunity.

## 4. Materials and Methods

### 4.1. Insect Rearing and Preparation of Microorganisms

*T. molitor* larvae were reared under dark conditions at 26 ± 1 °C and 60% ± 5% relative humidity in an environmental chamber established in our laboratory. Larvae were fed with an artificial diet consisting of 1.1 g sorbic acid, 1.1 mL propionic acid, 20 g bean powder, 10 g brewer’s yeast powder, and 200 g wheat bran in 4400 mL distilled water. The feed was autoclaved at 121 °C for 15 min. All experiments were conducted with healthy 10th–12th instar larvae.

Gram-negative *E. coli* (strain K12), Gram-positive *S. aureus* (strain RN4220), and fungus *C. albicans* (strain AUMC 13529) were used as representatives of different pathogenic sources to investigate the roles of *Tm*TLR2 in immunological challenges. *E. coli* and *S. aureus* were cultured in Luria–Bertani [13] broth, and *C. albicans* was cultured in Sabouraud’s dextrose broth at 37 °C overnight. The microorganisms were harvested, washed twice in 1× phosphate-buffered saline (PBS; 8.0 g NaCl, 0.2 g KCl, 1.42 g Na_2_HPO_4_, 0.24 g KH_2_PO_4_ in 1 L distilled water, pH 7.0), and centrifuged at 3500 rpm for 15 min. Subsequently, the samples were suspended in PBS, and concentrations were measured at 600 nm (OD600) through spectrophotometry (Eppendorf, Hamburg, Germany). *E. coli* and *S. aureus* were diluted to 1 × 10^6^ cells/µL, and *C. albicans* was diluted to 5 × 10^4^ cells/µL for immune challenge studies. The relevant optimization of microbial concentration was adjusted in accordance with our previous studies [20,53,54].

### 4.2. In Silico Analysis of TmToll-2 

The *T. castaneum* TLR2 amino acid sequence (accession number: XP_015837871.1) was used as a query to perform Local-tblastn analysis and obtain *TmToll-2* gene sequence (accession number: OP566501) from RNAseq analysis and NCBI Expressed Sequence Tag [28] database [55]. The full-length ORF and deduced amino acid sequences of *Tm*TLR2 were analyzed using BLASTp (NCBI; https://blast.ncbi.nlm.nih.gov/Blast.cgi). The multiple sequence alignment of the *Tm*TLR2 amino acid sequence with representative TLR amino acid sequences from other insects (retrieved from GenBank) was generated using ClustalX 2.1 [56]. A phylogenetic tree was constructed on the basis of amino acid sequence alignments via the maximum likelihood method (bootstrap trial set to 1000) with several protein sequences, including those of *Tc*Toll-2, *T. castaneum* toll-like receptor 2 (XP_015837871.1); *So*Toll-2, *S. oryzae* toll-like receptor 2 (XP_030759691.1); *Aa*Tollo, *Anopheles arabiensis* toll-like receptor Tollo (XP_040168234.1); *Dm*Toll-9XC, *D. melanogaster* Toll-9, isoform C (NP_001246846.1); *Ms*Toll-2, *M. sexta* toll-like receptor 2 (XP_030022594.2); *Vc*Toll-2, *V. cardui* toll-like receptor 2 (XP_046969474.1); *Ba*Toll-2, *Bicyclus anynana* toll-like receptor 2 (XP_023948156.1); *Gm*Toll-2, *Galleria mellonella* toll-like receptor 2 (XP_031764691.1); *Bm*Toll-4, *B. mori* toll-like receptor 4 (XP_012546905.2); *Nf*Toll-2X1, *Neodiprion fabricii* toll-like receptor 2 isoform X1 (XP_046422597.1); *Nl*Toll-2-T2-X2, *Nilaparvata lugens* toll-like receptor 2 type-2 isoform X1 (XP_039281792.1); *Hv*Toll-2-T2, *Homalodisca vitripennis* toll-like receptor 2 type-2 (XP_046658785.1); and MmToll-2X1, *M. musculus* toll-like receptor 2 isoform X1 (XP_006501523.1). Phylogenetic analyses were performed using the Tree Explorer view in Molecular Evolutionary Genetics Analysis (MEGA) version 7.0 [57] (https://megasoftware.net).

### 4.3. Expression and Induction Pattern Analysis of TmToll-2 

Total RNA was isolated at different developmental stages, including eggs, young larvae (instars 10–12), late larvae (instars 14–15), prepupae, 1-to-7-day-old pupae, and 1-to-5-day-old adults and tissues (integument (IT), gut, fat bodies (FBs), Malpighian tubules (MTs), hemocytes [28] of last instar larvae and 5-day-old adults, and ovary [17] and testis (TE) of 5-day-old adults) of *T. molitor* to investigate the temporal and spatial expression patterns of *TmToll-2*.

Suspensions containing 1 × 10^6^ cells/µL of *E. coli* and *S. aureus* and 5 × 10^4^ cells/mL of *C. albicans* were injected into *T. molitor* larvae at 10th–12th instars (n = 20) to analyze the induction pattern of *TmToll-2* against microorganisms. PBS-injected *T. molitor* larvae were used as the control group. Samples were collected 3, 6, 9, 12, and 24 h after the microbial challenge.

Total RNA was extracted by using a Clear-S total RNA extraction kit (Invirustech Co., Gwangju, Republic of Korea) in accordance with the manufacturer’s instructions. Then, 2 mg of total RNA was used as the template to synthesize cDNA by using the Oligo (dT)12–18 primers under the following reaction conditions: 72 °C for 5 min, 42 °C for 1 h, and 94 °C for 5 min. MyGenie96 Thermal Block (Bioneer, Daejeon, Republic of Korea) and AccuPower^®^ RT PreMix (Bioneer) were used in accordance with the manufacturer’s instructions. cDNA was stored at −20 °C until further use.

The relative mRNA expression level of *TmToll-2* was investigated via quantitative real-time polymerase chain reaction (qRT-PCR) by using an AccuPower^®^ 2X Greenstar^TM^ qPCR Master Mix (Bioneer, Daejeon, Republic of Korea) with synthesized cDNAs and specific primers (Table 1) at an initial denaturation of 95 °C for 5 min, followed by 45 cycles at 95 °C for 15 s, and 60 °C for 30 s. *T. molitor* ribosomal protein (*TmL27a*) was used as an internal control, and the results were analyzed using the 2^−ΔΔCt^ method [58]. The results were presented as means ± standard error (SE) of three biological replicates.

### 4.4. RNA Interference Analysis

*TmToll-2* gene silencing was performed as reported previously [20,37,59]. Briefly, the primers containing the T7 promoter sequence at their 5′ end were designed using SnapDragon-Long dsRNA Design (Table 1) to synthesize the dsRNA of the *TmToll-2* gene. The primary PCR for the *TmToll-2* gene was performed using an AccuPower PfuPCR PreMix (Bioneer) with cDNA and specific primers of the *TmToll-2* gene (Table 1). The second PCR was conducted with primers tailed with T7 promoter sequences and 100× dilution of the second PCR products. According to the developmental expression pattern of *TmToll-2*, cDNA was synthesized from the whole bodies of the pupae on day 5 and used as a template under the following cycling conditions: an initial denaturation step at 94 °C for 2 min followed by 35 cycles of denaturation at 94 °C for 30 s, annealing at 53 °C for 30 s, and extension at 72 °C for 30 s, with a final extension step at 72 °C for 5 min. The PCR products were purified using a Clear-STM PCR/Gel DNA fragment purification kit (Invirustech Co., Gwangju, Republic of Korea). The dsRNA was synthesized using an AmpliScribe T7-Flash transcription kit (Epicentre Biotechnologies, Madison, WI, USA) in accordance with the manufacturer’s instructions. After synthesis, the dsRNA was purified by precipitation with 5 M ammonium acetate and 80% ethanol and quantified using an Epoch spectrophotometer (BioTek Instruments, Inc., Winooski, VT, USA). Then, 2 µg of synthesized ds*TmToll-2* RNAi was injected into 10th–11th instar larvae for gene silencing, and ds*TmVer* was used as a control.

### 4.5. Effect of TmToll-2 Gene Silencing on Larval Mortality against Microbial Challenge

ds*TmToll-2* RNAi (2 µg/µL) was first injected into early-instar larvae (instars 10–12; n = 30) by using disposable needles mounted onto a micro-applicator (Picospritzer III Micro Dispense System; Parker Hannifin, Hollis, NH, USA) to measure the effect of *TmToll-2* in the *T. molitor* mortality following a pathogenic invasion. An equal amount of ds*TmVer* was injected into the larvae at the same stage as the negative control. The efficiency of *TmToll-2* knockdown was evaluated through qRT-PCR, and over 95% knockdown was achieved 4 days after injection. *TmToll-2*-silenced and ds*TmVer*-injected larval groups were challenged with *E. coli* (10^6^ cells/µL), *S. aureus* (10^6^ cells/µL), or *C. albicans* (5 × 10^4^ cells/µL) in triplicate experiments. The number of the surviving challenged larvae was monitored for 10 days, and the survival rates of *TmToll-2*-silenced larvae were compared with those of the control larvae. The relevant analysis was performed using Kaplan–Meier plots [60].

### 4.6. Effect of dsTmToll-2 RNAi on AMP and NF-κB Expression in Response to Microbial Challenge

The *TmToll-2* silenced larvae were challenged with *E. coli*, *S. aureus*, or *C. albicans* via RNAi technology to further evaluate the function properties of the *TmToll-2* gene in the humoral innate immune response. ds*TmVer* and PBS were used as the negative and injection controls, respectively. At 24 h post injection, the fat bodies, gut, and MTs were dissected, total RNA was extracted from each tissue, and cDNA was synthesized as described above. Subsequently, qRT-PCR was applied to study the expression levels of 15 AMP genes: *TmTenecin-1*, *-2*, *-3*, and *-4* (*TmTene1*, *2*, *3*, and *4*); *TmAttacin-1a*, *-1b*, and *-2* (*TmAtt1a*, *1b*, and *2*); *TmDefensin* (*TmDef*); *TmDefensin-like* (*TmDef-like*); *TmColeoptericin-A*, *-B*, and *-C* (*TmColeA*, *B*, and *C*); *TmCecropin-2* (*TmCec-2*); and *TmThaumatin like protein-1* and *-2* (*TmTLP1* and *2*). Moreover, the expression patterns of NF-κB genes, such as *TmDorsal* isoform *X1* and *X2* (*TmDorX1* and *X2*), and *TmRelish*, were investigated. A relative quantitative PCR was performed as mentioned above by using the AMP- and NF-κB gene-specific primers (Table 1). 

### 4.7. Statistical Analysis

All experiments were carried out in triplicate, and data were subjected to one-way ANOVA. Tukey’s multiple range tests were used to evaluate the difference between groups (*p* < 0.05). The fold change in gene expression levels compared with the internal control (*TmL27a*) and external control (PBS) levels was calculated using the 2^−ΔΔCt^ method. 

## Figures and Tables

**Figure 1 ijms-23-14490-f001:**
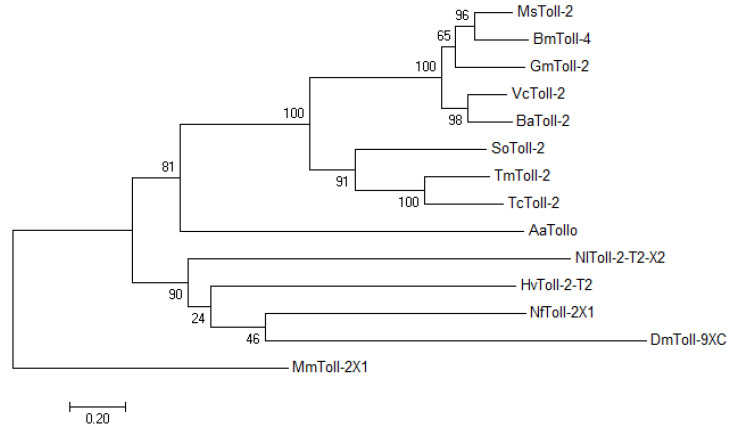
Molecular phylogenetic analysis of *TmToll-2*, *T. molitor Toll-2* (OP566501). The phylogenetic tree was constructed using MEGA7 with the maximum likelihood method and 1000 bootstrap replicates (where numbers at nodes indicate bootstrap support). The percentage of trees in which the associated taxa clustered together is shown next to the branches. A neighbor-joining (NJ) tree was constructed on the basis of the protein sequences of *Tc*Toll-2, *Tribolium castaneum* toll-like receptor 2 (XP_015837871.1); *So*Toll-2, *Sitophilus oryzae* toll-like receptor 2 (XP_030759691.1); *Aa*Tollo, *Anopheles arabiensis* toll-like receptor Tollo (XP_040168234.1); *Dm*Toll-9XC, *Drosophila melanogaster* Toll-9, isoform C (NP_001246846.1); *Ms*Toll-2, *Manduca sexta* toll-like receptor 2 (XP_030022594.2); *Vc*Toll-2, *Vanessa cardui* toll-like receptor 2 (XP_046969474.1); *Ba*Toll-2, *Bicyclus anynana* toll-like receptor 2 (XP_023948156.1); *Gm*Toll-2, *Galleria mellonella* toll-like receptor 2 (XP_031764691.1); *Bm*Toll-4, *Bombyx mori* toll-like receptor 4 (XP_012546905.2); *Nf*Toll-2X1, *Neodiprion fabricii* toll-like receptor 2 isoform X1 (XP_046422597.1); *Nl*Toll-2-T2-X2, *Nilaparvata lugens* toll-like receptor 2 type-2 isoform X1 (XP_039281792.1); *Hv*Toll-2-T2, *Homalodisca vitripennis* toll-like receptor 2 type-2 (XP_046658785.1); and MmToll-2X1, *Mus musculus* toll-like receptor 2 isoform X1 (XP_006501523.1), which was used as the outgroup.

**Figure 2 ijms-23-14490-f002:**
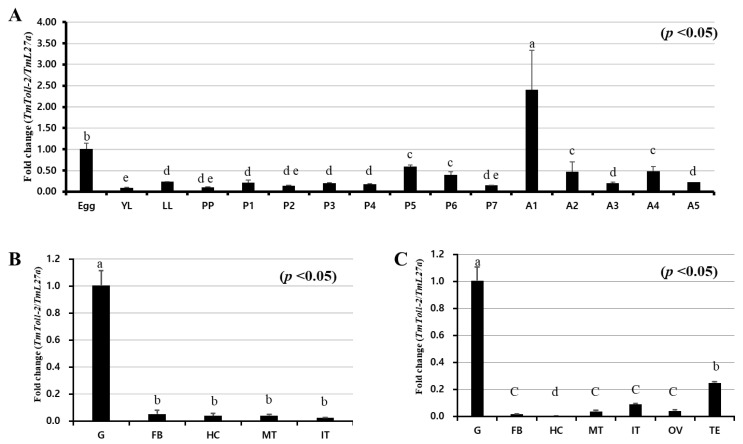
Developmental stage- and tissue-specific expression patterns of *TmToll-2* measured by qRT-PCR. (**A**) Relative mRNA expression levels of *TmToll-2* in eggs, young larvae, late-instar larvae (LL), pre-pupae, 1-to-7-day-old pupae (P1–P7), and 1-to-5-day-old adults (A1–A5) were presented. Expression levels were the highest in the eggs and 1-day-old adults. The mRNA expression decreased at the larval stage and were the lowest at the young larval stage. *TmToll-2* tissue expression patterns in late instar larvae (**B**) and adults (**C**) were also examined. Total RNA was extracted from different tissues, including the integument (IT), Malpighian tubule (MT), gut (GT), hemocytes, and fat bodies (FB) of late instar larvae and the IT, MT, GT, hemocytes, FB, ovary, and testis (TE) of 5-day-old adults. Total RNA was isolated from 20 mealworms, and *T. molitor* 60S ribosomal protein 27a (*TmL27a*) primers were used as internal control (N = 3). One-way ANOVA and Tukey’s multiple-range test were used for comparisons. Bars with the same letter are not significantly different by Tukey’s multiple-range test (*p* < 0.05).

**Figure 3 ijms-23-14490-f003:**
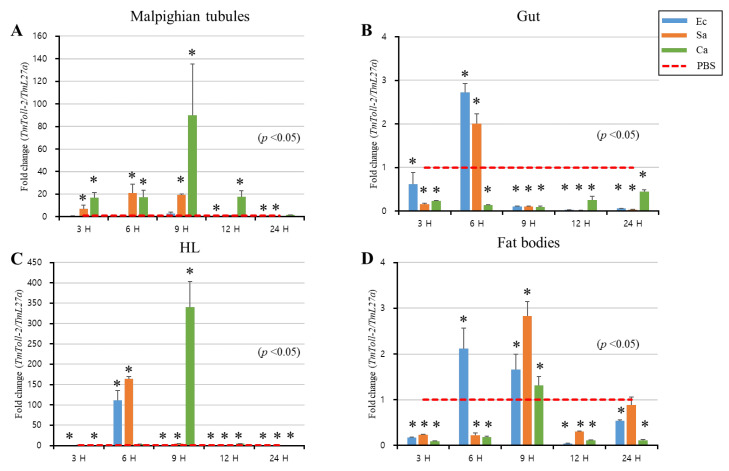
mRNA expression patterns of *TmToll-2* in immune-challenged *T. molitor* larvae. mRNA levels of *TmToll-2* in the Malpighian tubules (**A**), gut (**B**), hemocytes (**C**), and fat bodies (**D**) were examined by qRT-PCR 3, 6, 9, 12, and 24 h after infection with *E. coli* (10^6^ cells/µL), *S. aureus* (10^6^ cells/µL), and *C. albicans* (5 × 10^4^ cells/µL). *TmToll-2* mRNA expression was upregulated in response to all infectious sources and varied in tissue- and time-dependent manners. The upregulation of *TmToll-2* gene expression was the highest in the hemolymph and MTs against *C. albicans*. PBS was used as an injection control, and *T. molitor* 60S ribosomal protein 27a (*TmL27a*) primers were used as internal control (n = 3). Asterisks indicate significant differences between infected and PBS-injected larval groups by Student’s *t*-test (*p* < 0.05). Vertical bars indicate means ± SD (n = 20).

**Figure 4 ijms-23-14490-f004:**
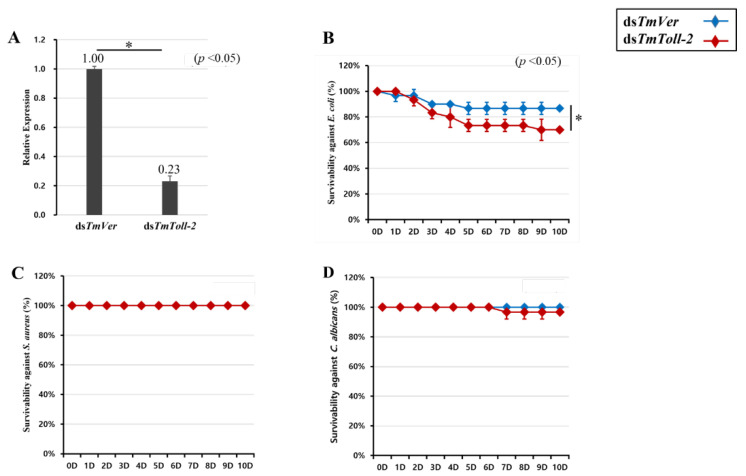
Effect of *TmToll-2* gene silencing on *T. molitor* larval survival. The RNAi efficiency of ds*TmToll-2* RNAi was measured by qRT-PCR 4 days after injection (**A**). *TmToll-2*-silenced larvae were injected with *E. coli* (**B**), *S. aureus* (**C**), and *C. albicans* (**D**), and survival rates were studied for 10 days post-pathogen injection (n = 10 per group). Larval survival rates at 10 days post-microbial injection were 80% after *E. coli* injection and 100% after *S. aureus* and *C. albicans* injection compared with the levels in the ds*TmVer*-injected control group. Data were reported as averages of three biologically independent replicates. Asterisks indicate significant differences between ds*TmToll-2*- and ds*TmVer* RNAi-injected groups. Survival analysis was performed using Kaplan–Meier plots (log-rank chi-squared test; * *p* < 0.05).

**Figure 5 ijms-23-14490-f005:**
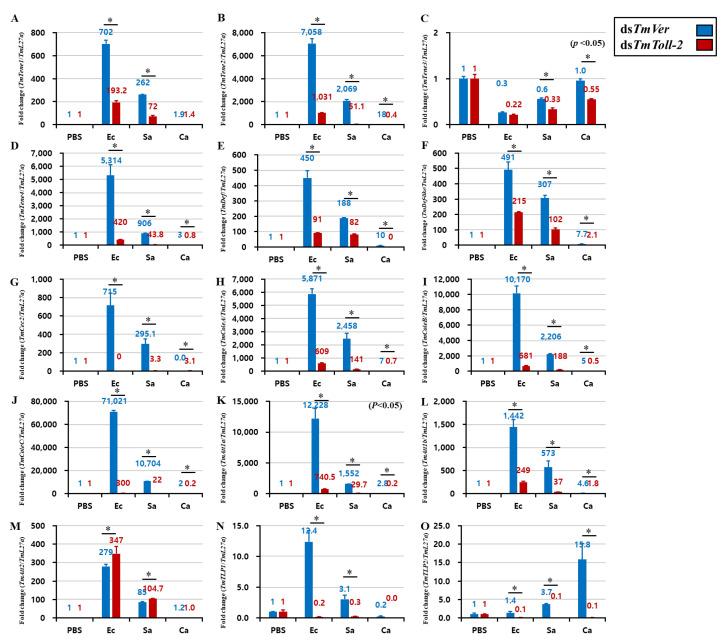
Induction of 15 AMP genes in the fat bodies of *TmToll-2*-treated *T. molitor* larvae infected with *E. coli* (Ec), *S. aureus* (Sa), and *C. albicans* (Ca) by using PBS as a control. At 24 h post microbial injection, AMP genes, including *TmTene1* (**A**), *TmTene2* (**B**), *TmTene3* (**C**), *TmTene4* (**D**), *TmDef* (**E**), *TmDef-like* (**F**), *TmCec2* (**G**), *TmColeA* (**H**), *TmColeB* (**I**), *TmColeC* (**J**), *TmAtt1a* (**K**), *TmAtt1b* (**L**), *TmAtt2* (**M**), *TmTLP1* (**N**), and *TmTLP2* (**O**) were examined via qPCR by using ds*TmVer* as a knockdown control and *T. molitor* ribosomal protein (*TmL27a*) as an internal control. All experiments were performed in triplicate. Asterisks indicate significant differences between ds*TmToll-2*- and ds*TmVer* RNAi-treated groups determined by Student’s *t*-test (*p* < 0.05).

**Figure 6 ijms-23-14490-f006:**
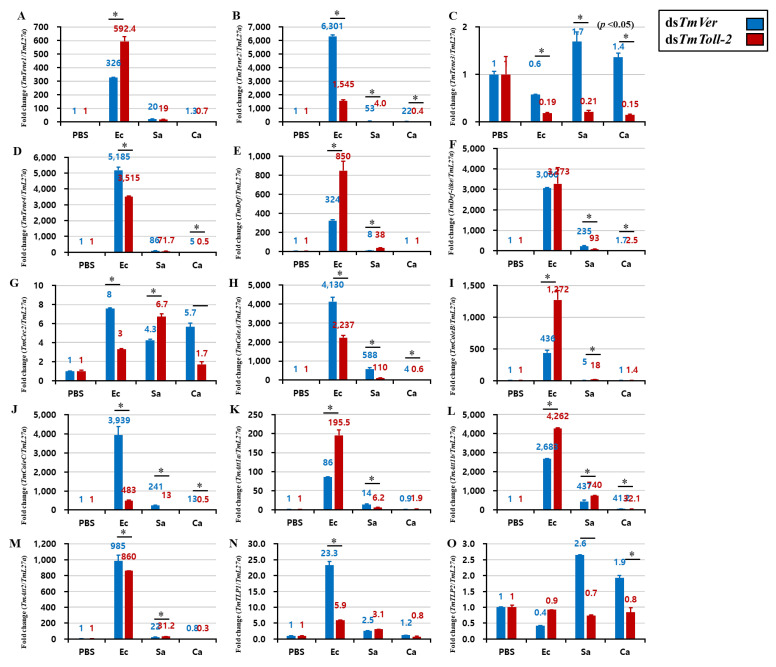
Effect of *TmToll-2* RNAi on the induction of 15 AMP genes in the gut of *T. molitor* larvae infected with *E. coli* (Ec), *S. aureus* (Sa), and *C. albicans* (Ca) by using PBS as the control. At 24 h post microbial injection, AMP genes, including *TmTene1* (**A**), *TmTene2* (**B**), *TmTene3* (**C**), *TmTene4* (**D**), *TmDef* (**E**), *TmDef-like* (**F**), *TmCec2* (**G**), *TmColeA* (**H**), *TmColeB* (**I**), *TmColeC* (**J**), *TmAtt1a* (**K**), *TmAtt1b* (**L**), *TmAtt2* (**M**), *TmTLP1* (**N**), and *TmTLP2* (**O**) were examined via qPCR by using ds*TmVer* as a knockdown control and *T. molitor* ribosomal protein (*TmL27a*) as an internal control. All experiments were performed in triplicate. Asterisks indicate significant differences between ds*TmToll-2*- and ds*TmVer* RNAi-treated groups determined by Student’s *t*-test (*p* < 0.05).

**Figure 7 ijms-23-14490-f007:**
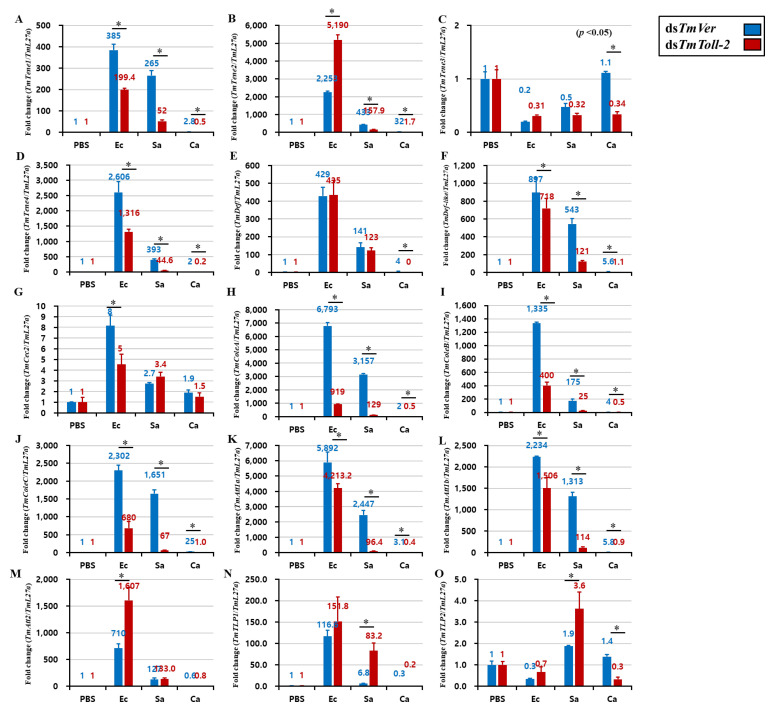
Effect of *TmToll-2* gene silencing on the induction of 15 AMP genes in the Malpighian tubules of *T. molitor* larvae infected with *E. coli* (Ec), *S. aureus* (Sa), and *C. albicans* (Ca) by using PBS as the control. At 24 h post microbial injection, AMP genes, including *TmTene1* (**A**), *TmTene2* (**B**), *TmTene3* (**C**), *TmTene4* (**D**), *TmDef* (**E**), *TmDef-like* (**F**), *TmCec2* (**G**), *TmColeA* (**H**), *TmColeB* (**I**), *TmColeC* (**J**), *TmAtt1a* (**K**), *TmAtt1b* (**L**), *TmAtt2* (**M**), *TmTLP1* (**N**), and *TmTLP2* (**O**) were examined via qPCR by using ds*TmVer* as a knockdown control and *T. molitor* ribosomal protein (*TmL27a*) as an internal control. All experiments were performed in triplicate. Asterisks indicate significant differences between ds*TmToll-2*- and ds*TmVer* RNAi-treated groups determined by Student’s *t*-test (*p* < 0.05).

**Figure 8 ijms-23-14490-f008:**
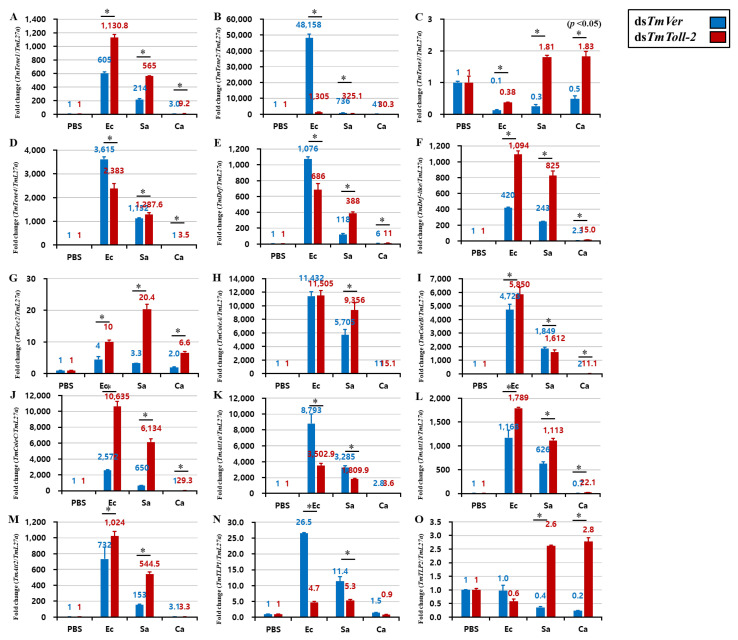
Induction of 15 AMP genes in the whole body of *TmToll-2*-silenced *T. molitor* larvae infected with *E. coli* (Ec), *S. aureus* (Sa), and *C. albicans* (Ca) by using PBS as control. At 24 h post microbial injection, AMP genes, including *TmTene1* (**A**), *TmTene2* (**B**), *TmTene3* (**C**), *TmTene4* (**D**), *TmDef* (**E**), *TmDef-like* (**F**), *TmCec2* (**G**), *TmColeA* (**H**), *TmColeB* (**I**), *TmColeC* (**J**), *TmAtt1a* (**K**), *TmAtt1b* (**L**), *TmAtt2* (**M**), *TmTLP1* (**N**), and *TmTLP2* (**O**) were examined via qPCR by using ds*TmVer* as a knockdown control and *T. molitor* ribosomal protein (*TmL27a*) as an internal control. All experiments were performed in triplicate. Asterisks indicate significant differences between ds*TmToll-2*- and ds*TmVer* RNAi-treated groups determined by Student’s *t*-test (*p* < 0.05).

**Figure 9 ijms-23-14490-f009:**
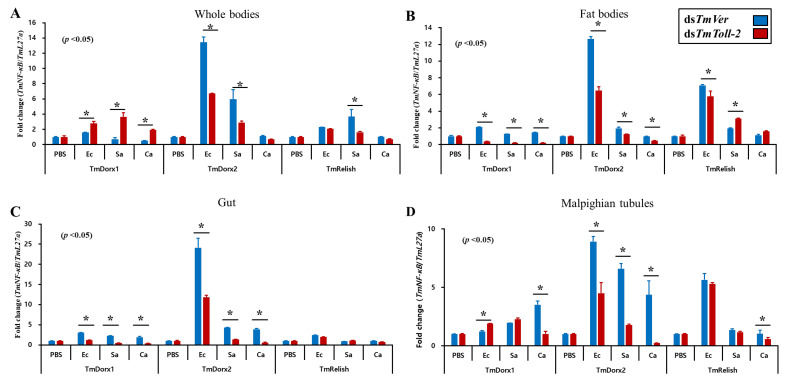
Effect of *TmToll-2* gene silencing on NF-kB gene expression. ds*TmToll-2* RNAi-treated *T. molitor* larvae were infected with *E. coli*, *S. aureus*, and *C. albicans*; at 24 h post pathogen injection, the mRNA levels of the NF-kB pathway genes *TmDorX1*, *TmDorX2*, and *TmRelish* in the whole body (**A**), fat bodies (**B**), gut (**C**), and Malpighian tubules (**D**) were measured via RT-qPCR. The expression level of *TmDorx2* was suppressed following *E. coli* and *S. aureus* infection in all the examined tissues, while the level of *TmRelish* was not affected. In the fat bodies and gut, the NF-κB genes expression was negatively regulated. *TmVer* dsRNA was assessed as a negative control, and *T. molitor* ribosomal protein (*TmL27a*) was used as an internal control. All experiments were performed in triplicate. Asterisks indicate significant differences between ds*TmToll-2*- and ds*TmVer* RNAi-treated groups determined using Student’s *t*-test (*p* < 0.05).

**Table 1 ijms-23-14490-t001:** Primers used in the present study.

Name	Primer Sequences (5′- → -3′)
TmToll-2-qPCR-Fw	TCTAGTAGACGTAGCGGTGA
TmToll-2-qPCR-Rev	AATCGCAAGTGAATGGGTTG
TmToll-2-T7-Fw	TAATACGACTCACTATAGGGTTCGGCGAAGACAAAGAAAGT
TmToll-2-T7-Rev	TAATACGACTCACTATAGGGTCCAAACCATCAAAACATCCC
TmL27a-qPCR-Fw	TCATCCTGAAGGCAAAGCTCCAGT
TmL27a-qPCR-Rev	AGGTTGGTTAGGCAGGCACCTTTA
TmVer-T7-Fw	TAATACGACTCACTATAGGGTCGAGAAGTCAGAGCAGCAA
TmVer-T7-Rev	TAATACGACTCACTATAGGGTACCACCAGTTCCCAGTTGAG
TmTenecin-1-Fw	CAGCTGAAGAAATCGAACAAGG
TmTenecin-1-Rev	CAGACCCTCTTTCCGTTACAGT
TmTenecin-2_Fw	CAGCAAAACGGAGGATGGTC
TmTenecin-2-Rev	CGTTGAAATCGTGATCTTGTCC
TmTenecin-3-Fw	GATTTGCTTGATTCTGGTGGTC
TmTenecin-3-Rev	CTGATGGCCTCCTAAATGTCC
TmTenecin-4-Fw	GGACATTGAAGATCCAGGAAAG
TmTenecin-4-Rev	CGGTGTTCCTTATGTAGAGCTG
TmDefensin-Fw	AAATCGAACAAGGCCAACAC
TmDefensin-Rev	GCAAATGCAGACCCTCTTTC
TmDefensin-like-Fw	GCGATGCCTCATGAAGATGTAG
TmDefensin-like-Rev	CCAATGCAAACACATTCGTC
TmColoptericinA-Fw	GGACAGAATGGTGGATGGTC
TmColoptericinA-Rev	CTCCAACATTCCAGGTAGGC
TmColoptericinB-Fw	CAGCTGTTGCCCACAAGTG
TmColoptericinB-Rev	CTCAACGTTGGTCCTGGTGT
TmColoptericinC-Fw	CAGCTGTTGCCCACAAGTG
TmColoptericinC-Rev	CTCAACGTTGGTCCTGGTGT
TmAttacin-1a-Fw	GAAACGAAATGGAAGGTGGA
TmAttacin-1a-Rev	TGCTTCGGCAGACAATACAG
TmAttacin-1b-Fw	CCCTCTGATGAAACCTCCAA
TmAttacin-1b-Rev	GAGCTGTGAATGCAGGACAA
TmAttacin-2-Fw	AACTGGGATATTCGCACGTC
TmAttacin-2-Rv	CCCTCCGAAATGTCTGTTGT
TmCecropin-2-Fw	TACTAGCAGCGCCAAAACCT
TmCecropin-2-Rev	CTGGAACATTAGGCGGAGAA
TmThaumatin-likeprotein-1-Fw	CTCAAAGGACACGCAGGACT
TmThaumatin-like protein-1-Rev	ACTTTGAGCTTCTCGGGACA
TmThaumatin-like protein-2-Fw	CCGTCTGGCTAGGAGTTCTG
TmThaumatin-like protein-2-Rev	ACTCCTCCAGCTCCGTTACA
TmDorsal1-qPCR-Fw	AGCGTTGAGGTTTCGGTATG
TmDorsal1-qPCR-Rev	TCTTTGGTGACGCAAGACAC
TmDorsal2-qPCR-Fw	ACACCCCCGAAATCACAAAC
TmDorsal2-qPCR-Rev	TTTCAGAGCGCCAGGTTTTG
TmRelish-qPCR-Fw	AGCGTCAAGTTGGAGCAGAT
TmRelish-qPCR-Rev	GTCCGGACCTCATCAAGTGT

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
