# Peer review of "Immunological Roles of TmToll-2 in Response to Escherichia coli Systemic Infection in Tenebrio molitor"

_ijms, 2022, doi:10.3390/ijms232214490_

Round 1
Reviewer 1 Report
These experiments are potentially interesting. However, several descriptions seem to be immature and poor. The manuscript should be revised extensively.
Specific comments:
1. Descriptions and terms were confused throughout manuscript between mammals and insects. For example, Toll-like receptor (TLR) and NF-κB are mammals ones, not insect ones. Please distinguish accurately.
2. Usage of mRNAi names were confusable. For example, the double-stranded TmToll-2-injected and dsTmToll-2. “RNAi” should be added after these words.
3. The authors denoted mRNA expression levels as fold change. What do yo mean “fold change”? What is a control as 1.0?
4. Line 55: “, and virulence factors” cannot be understand.
5. Line 144-145: TmToll-2 gene silencing significantly suppressed E. coli-induced larvae death. This description might be wrong.
6. Figure 3: How about mRNA levels at time zero, namely untreated control. In ML and HL, bars of some expression levels (fold change?), which were significantly different, cannot be seen because the values were very low. Logarithmic scale would be better.
7. Figure 3: In gut and fat bodies, almost mRNA expression levels (fold change?) were less than 1. This indicated that the mRNA levels of TmToll-2 were suppressed by the stimulation with bacteria/fungi, didn’t it?
8. Figure 4-C: P<0.05 seems to be wrong.
9. Figure 5, and 6: In the legends, “gene silencing” should be described clearly.
Author Response
Rebuttal letter to reviewer’s comments on our submitted manuscript (1981714)
We received the reviewers comment for the submitted manuscript (1981714). We are sincerely grateful for the reviewers’ positive and constructive comments, which have been very helpful to improve the quality of our paper. The suggestions have noted and we have worked on the reviewer comments. We have taken care of all comments and manuscript has been edited based on the comments. We hope the manuscript has improved with the suggested revisions and would be a novel statement to the scientific community covered by the journal.
Please find the author’s comments to reviewers’ queries as under:
Independent Review Report:
Reviewer #1 (Comments to the Author):
These experiments are potentially interesting. However, several descriptions seem to be immature and poor. The manuscript should be revised extensively.
Specific comments:
- Descriptions and terms were confused throughout manuscript between mammals and insects. For example, Toll-like receptor (TLR) and NF-κB are mammals ones, not insect ones. Please distinguish accurately.
Authors Response: We own the pleasure of receiving your constructive comments on this manuscript. Regarding the inquires mentioned above; please note that as it has been mentioned in the introduction, Toll receptor has been identified for the first time in Drosophila therefore this receptor in the other organisms from vertebrates (including mammals) to invertebrates (including beetles) is called Toll-like receptors. Moreover, please note that NF-κB transcription factors are used by the same terminology for boths mammals and insects.
- Usage of mRNAi names were confusable. For example, the double-stranded TmToll-2-injected and dsTmToll-2. “RNAi” should be added after these words.
Authors Response: We appreciate your concern regarding the referred issue. Therefore, we revised our manuscript according to your notice.
- The authors denoted mRNA expression levels as fold change. What do yo mean “fold change”? What is a control as 1.0?
Author’s response: We are grateful for your fine consideration. Please be informed that fold changes referred to mRNA expression level changes monitored during qPCR cycles compared to the mRNA expression of L27a (housekeeping gene in the same sample) and asterisks indicate significant differences between infected and PBS-injected larval (control) groups by Student’s t-test.
- Line 55: “, and virulence factors” cannot be understand.
Author’s response: Thank you for another fine notice. Please note that Virulence factors are the molecules that assist the bacterium colonize the host at the cellular level. These factors are either secretory, membrane associated or cytosolic in nature. The cytosolic factors facilitate the bacterium to undergo quick adaptive—metabolic, physiological and morphological shifts.
- Line 144-145: TmToll-2 gene silencing significantly suppressed E. coli-induced larvae death. This description might be wrong.
Author’s response: Thank you for this fine notice. Required revision has been done according to your sensible point and mentioned sentence has been changed.
- Figure 3: How about mRNA levels at time zero, namely untreated control. In ML and HL, bars of some expression levels (fold change?), which were significantly different, cannot be seen because the values were very low. Logarithmic scale would be better.
Author’s response: We appreciate your concern. Please note that mRNA expression in none of the dissected tissues has been checked at time zero. However, at the earliest time point (3 hours) post infection the mRNA expression of TmToll-2 was reported to be very low. As it was mentioned in your sensible comment, we can make a bar plot that breaks extremely large bars. However, please bear in mind that the relevant expression in MTs and Hl was negligible to be bolded in the sense that the focal point of this graph is the high expression of TmToll-2 following the infection and pathogen specificity of the relevant expression. Therefore, we believe that broken bar plots may cause confusion for the readers.
- Figure 3: In gut and fat bodies, almost mRNA expression levels (fold change?) were less than 1. This indicated that the mRNA levels of TmToll-2 were suppressed by the stimulation with bacteria/fungi, didn’t it?
Author’s response: We appreciate your frequent sensible points. However, kindly mark that as it has been described within discussion section, seven TLR has been identified in T. molitor and low expression following infection in different time points and different distributions of different TLR in these tissue can be also related to gene redundancy and homeostasis in T. molitor related to other isoforms of this gene or cross regulation with other signaling pathways involve in innate immunity.
- Figure 4-C: P<0.05 seems to be wrong.
Author’s response: Thank you for another fine notice. Kindly mark that mentioned figured has been revised according to your comment.
- Figure 5, and 6: In the legends, “gene silencing” should be described clearly.
Author’s response: We are deeply thankful for your sharp and constructive notices. With reference to the to the above-mentioned comment kindly note that figure 5 to 9 are related to AMP and NF-κB transcription factors mRNA expression, therefore, further description related to gene silencing shall be considered as irrelevant and confusing to the readers. However, the relevant explanation related to gene silencing experiments has been mentioned in figure4 legend.

Reviewer 2 Report
Manuscript ID: IJMS-1981714
Review Report
Title: Immunological roles of TmToll-2 in response to Escherichia coli systemic infection in Tenebrio molitor
This study is focusing on the antimicrobial effect of TmToll-2 at the larval stage of Mealworm (Tenebrio molitor) against E. coli, Staphylococcus aureus and candida albicans. Knockdown of the TmToll-2 gene by using RNAi technology followed by dsTmToll-2 injection in T. molitor larvae. Silencing of tmToll-2 gene caused mortality in larvae after infection with E.coli. The techniques used in this study are the growth of bacteria, RNA extraction, gene silencing and challenge study with different microbes and gene expression by real-time PCR (q-PCR). The study is very convincing. The manuscript can be accepted with minor revisions.
1. The study has used larval stages of mealworms for infection and gene expression using different organs. I recommend the author draw the figure of the complete study design by including a picture of different larval stages of mealworms with different organs. By drawing such figure will make better readability and it will simplify to follow the complete manuscript.
2. Ethical approval: Author did not mention the ethical approval before infection with pathogenic microbes in Mealworm. This study used GMO (dsTmTOLL-2) and pathogenic bacteria E. coli K12, S. aureus RN4220 and C. albicans AUMC13529 for vaccination and challenge study. I suggest that Author should mention the ethical approval in the manuscript.
3. Please make sure that the TmToll-2 should be written in the same way throughout the manuscript. I observed that author has use TmTLR13 (line no-307), tmTLR2 (line no.323)
4. Author should mention the annealing temperature of each primer used for q-PCR (Table-1) and mention the source of primers or NCBI accession number from where it was designed.

Author Response
Rebuttal letter to reviewer’s comments on our submitted manuscript (1981714)
We received the reviewers comment for the submitted manuscript (1981714). We are sincerely grateful for the reviewers’ positive and constructive comments, which have been very helpful to improve the quality of our paper. The suggestions have noted and we have worked on the reviewer comments. We have taken care of all comments and manuscript has been edited based on the comments. We hope the manuscript has improved with the suggested revisions and would be a novel statement to the scientific community covered by the journal.
Please find the author’s comments to reviewers’ queries as under:
Independent Review Report:
Reviewer #2 (Comments to the Author):
This study is focusing on the antimicrobial effect of TmToll-2 at the larval stage of Mealworm (Tenebrio molitor) against E. coli, Staphylococcus aureus and candida albicans. Knockdown of the TmToll-2 gene by using RNAi technology followed by dsTmToll-2 injection in T. molitor larvae. Silencing of tmToll-2 gene caused mortality in larvae after infection with E.coli. The techniques used in this study are the growth of bacteria, RNA extraction, gene silencing and challenge study with different microbes and gene expression by real-time PCR (q-PCR). The study is very convincing. The manuscript can be accepted with minor revisions.
- The study has used larval stages of mealworms for infection and gene expression using different organs. I recommend the author draw the figure of the complete study design by including a picture of different larval stages of mealworms with different organs. By drawing such figure will make better readability and it will simplify to follow the complete manuscript.
Author’s response: We are deeply grateful for your constructive and fine comments. Please note that according to your suggestion we have prepared a figure (Supplementary) as our experimental research procedures to simplify our study for the readers.
Supplementary Figure 1.
- Ethical approval: Author did not mention the ethical approval before infection with pathogenic microbes in Mealworm. This study used GMO (dsTmTOLL-2) and pathogenic bacteria E. coli K12, S. aureus RN4220 and C. albicans AUMC13529 for vaccination and challenge study. I suggest that Author should mention the ethical approval in the manuscript.
Author’s response: We are grateful for your fine consideration. However, please be informed that according to the Principles of Humane Experimental Technique (Russell and Burch, 1959), arbitrarily excluded invertebrates from humane consideration. Kindly note that ethics approval is not required for experiments involving invertebrates (e.g. mealworms, earthworms, waxworms, silkworms, fruit flies) or single-celled micro-organisms (e.g. diatoms).
- Please make sure that the TmToll-2 should be written in the same way throughout the manuscript. I observed that author has use TmTLR13 (line no-307), tmTLR2 (line no.323)
Author’s response: Thank you for another fine notice. We have revised the mentioned sentences according to your sensible comment. Moreover, please note that TLR is referenced to the receptor protein and Toll is being used for the gene of the relevant protein.
- Author should mention the annealing temperature of each primer used for q-PCR (Table-1) and mention the source of primers or NCBI accession number from where it was designed.
Author’s response: Thank you for all your fine critics on this manuscript including the above matter. Kindly note that annealing temperature of the primers mentioned is Table 1 has been mentioned in the material and method section (Line 381) “an initial denaturation step at 94°C for 2 min followed by 35 cycles of denaturation at 94°C for 30 s, annealing at 53°C for 30 s, and extension at 72°C for 30 s, with a final extension step at 72°C for 5 min” and adding this particular information in the Table of primer sequence information shall be considered as irrelevant.

Round 2
Reviewer 1 Report
Thank you for your replies to my comments.
All replies are suffcient for my quenstions.